# High Diversity of Microcystin Chemotypes within a Summer Bloom of the Cyanobacterium *Microcystis botrys*

**DOI:** 10.3390/toxins11120698

**Published:** 2019-12-01

**Authors:** Emma Johansson, Catherine Legrand, Caroline Björnerås, Anna Godhe, Hanna Mazur-Marzec, Torbjörn Säll, Karin Rengefors

**Affiliations:** 1Department of Biology, Lund University, Ecology Building, Sölvegatan 35-37, 22362 Lund, Sweden; caroline.bjorneras@biol.lu.se (C.B.); torbjorn.sall@biol.lu.se (T.S.); 2Faculty of Health and Life Sciences, Linnaeus University, 39182 Kalmar, Sweden; catherine.legrand@lnu.se; 3Department of Marine Sciences, University of Gothenburg, Box 461, 40530 Göteborg, Sweden; 4Department of Marine Biotechnology, University of Gdansk, Marszałka J. Piłusudskiego 46, 81378 Gdynia, Poland; hanna.mazur-marzec@ug.edu.pl

**Keywords:** microcystin, *Microcystis botrys*, chemotypes, cyanobacteria, diversity

## Abstract

The fresh-water cyanobacterium *Microcystis* is known to form blooms world-wide, and is often responsible for the production of microcystins found in lake water. Microcystins are non-ribosomal peptides with toxic effects, e.g. on vertebrates, but their function remains largely unresolved. Moreover, not all strains produce microcystins, and many different microcystin variants have been described. Here we explored the diversity of microcystin variants within *Microcystis botrys*, a common bloom-former in Sweden. We isolated a total of 130 strains through the duration of a bloom in eutrophic Lake Vomb, and analyzed their microcystin profiles with tandem mass spectrometry (LC-MS/MS). We found that microcystin producing (28.5%) and non-producing (71.5%) *M. botrys* strains, co-existed throughout the bloom. However, microcystin producing strains were more prevalent towards the end of the sampling period. Overall, 26 unique *M. botrys* chemotypes were identified, and while some chemotypes re-occurred, others were found only once. The *M. botrys* chemotypes showed considerable variation both in terms of number of microcystin variants, as well as in what combinations the variants occurred. To our knowledge, this is the first report on microcystin chemotype variation and dynamics in *M. botrys*. In addition, our study verifies the co-existence of microcystin and non-microcystin producing strains, and we propose that environmental conditions may be implicated in determining their composition.

## 1. Introduction

Freshwater environments are of great importance as natural resources of drinking water, fish production, and recreation. Consequently, maintaining their quality [1] and safety is critical to society. Worldwide, cyanobacterial harmful algal blooms (CHABs) are increasing in many freshwater systems [2]. CHABs can negatively affect the ecology and environment of aquatic systems, and cause intoxication of livestock, pets, and humans [3]. Future scenarios predict that climate change will favor bloom-forming cyanobacteria, and that harmful blooms will increase both in frequency and duration, due to direct and indirect effects of changes in hydrological cycling, increased water temperatures, and nutrient loading [4,5,6,7,8,9].

In addition to forming dense blooms which can deteriorate water quality per se, many cyanobacterial genera produce secondary metabolites with significant bioactivity and toxicity to aquatic organisms and humans [10,11,12,13,14]. In freshwaters, microcystins are among the most common cyanotoxins [7,15]. These inhibit the eukaryotic protein phosphatases 1 and 2a, and have hepatotoxic effects in vertebrates [10,16,17]. Drinking water from surface water supplies must therefore be checked for microcystin content, and may not be higher than 1 µg microcystin L^−1^ according to WHO regulation. Microcystins are cyclic heptapeptides (Figure 1) and are produced by several cyanobacteria including *Microcystis*, *Planktothrix*, and *Anabaena* (*Dolichospermum*) [14,18,19]. The common structure and biosynthetic pathway of microcystin synthesis in these genera have been described [20,21,22,23,24,25]. In *Microcystis*, the microcystin synthetase gene cluster codes for a hybrid polyketide synthase (PKS)/non-ribosomal peptide synthetase (NRP) enzyme complex, and spans across 55 kb [22,23]. More than 240 microcystin variants have been described in microcystin producing cyanobacteria [26], and contribute to the high diversity of chemotypes reported in *Microcystis* [27]. The large amount of different structural variants is partly due to variable amino acids in position 2 and 4 in the cyclic structure, but there can also be variations in the other positions [20]. Individual strains have been shown to produce on average four different microcystins, but as many as 27 have been reported [28]. In addition, many cyanobacterial strains do not produce microcystins at all. Several studies (e.g., [29,30]) have shown that in natural populations, strains that produce microcystins co-exist with those that do not. These studies have been based on quantitative PCR targeted at one of the *mcy* genes, from which proportions of microcystin and non-microcystin producing strains have been inferred.

What factors regulate microcystin production, and the potential biological and/or ecological function of microcystins, has not yet been resolved [2]. Cyanotoxin production has been suggested to have ecological functions (e.g., predator protection, allelopathy) [31] but their long evolutionary history rather indicates that they are of importance for cellular processes, such as protein binding, photosynthesis, and growth [32,33,34]. In fact, both potential cellular and ecological functions of microcystins, such as protection against oxidative stress, predator defense, and increased competitive ability, have been shown [32,35,36]. Moreover, several studies (reviewed in [31]) indicate that microcystin production, and the level of toxicity in blooms, are influenced by environmental conditions, but also by the indirect effects of environment on cellular division and growth [31,37]. Furthermore, it has been shown that there is considerable temporal and spatial variation in microcystin production, and that microcystin variants vary in cytotoxicity [17,38,39,40]. Altogether this illustrates that the function of microcystin variants as well as their dynamics and distribution is not fully understood, and warrants further investigation.

Lake Vomb is a eutrophic lake in Skåne, southern Sweden. It serves as a drinking water supply for several municipalities, adding up to 500,000 consumers. Mass proliferations of *Microcystis spp.* have been reported frequently in Lake Vomb, as well as the occurrence of toxic blooms [41]. In this lake, *Microcystis botrys* is the most common species reported within the genus. However, we note that based on colony morphology it can sometimes be difficult to separate *M. botrys* from the closely related *M. aeruginosa.* Thus, it is possible that observations cited in the literature as *M. aeruginosa* may in fact be *M. botrys.* Frequent occurrence of *M. botrys* has been correlated to high levels of microcystin in bulk water samples both in Lake Vomb and other lakes in the drainage area [42,43]. A mesocosm experiment (with sediment and water sampled from regional lakes) showed that microcystin concentrations increased as the bloom progressed and peaked in late summer, coinciding with *M. botrys* peak biomass [42]. Thus, on a regional scale *M. botrys* is the dominant species in cyanobacterial blooms, and therefore of critical interest for drinking water safety.

The specific objectives of this study were to investigate the temporal dynamics of microcystin-producing (abbr. MC-producing) and non-microcystin-producing (abbr. non-MC-producing) *M. botrys* strains in Lake Vomb, and to gain insights into the variation of microcystin chemotypes during a bloom. In order to identify the specific microcystin variants present in the population and to explore the chemotype diversity among *M. botrys* strains, we used an approach based on isolating colonies and culturing, followed by tandem mass spectrometry LC-MS/MS of cell extracts. By culturing multiple individual strains it was possible to determine the proportion of MC- and non-MC-producing strains directly, without inferring from the presence of *mcy*-genes.

## 2. Results

In total, 843 individual *Microcystis* colonies were isolated from Lake Vomb during five sampling occasions (between 30 June and 8 September 2014). Colonies were monitored by microscopy regularly, and isolates contaminated with other algae were discarded. Of the initial isolates, 20% of the strains survived to established clonal cultures (Table 1). Of these, a total of 130 strains were analyzed for microcystin content by LC-MS/MS. In total, 37 of the 130 (28.5%) microcystin-profiled strains produced microcystins, and a total of 18 different microcystin variants were identified (Table 2).

The ratio between MC-producing and non-MC-producing strains changed during the sampling period, with a significant increase in MC-producing strains in late summer (chi-square test of goodness-of-fit: *X^2^* = 19,17, df = 4, *p*-value = 0.0007). A permutation test of equality of proportion gave a p-value of 0.00042, and a permutation to test for linear trend gave a p-value of <0.000001 based on a million runs, showing that the observed differences and trends of changes in proportions are significant, and were not due to chance. Among the 130 strains that were analyzed with LC-MS/MS, the presence of MC-producing strains was below 20% from June to early August (sampling occasions S1 20%; S2 8% and S3 15%), and increased during late August-early September (sampling occasions S4 45% and S5 52%) (Figure 2). In addition to the seasonal variation in the proportion of MC-producing strains, the mean number of different MC variants per MC-producing strain varied strongly among the sampling occasions. The means were 9.2, 1.0, 1.8, 3.9, and 6.3 for strains sampled during S1 to S5 respectively. The overall mean was 4.9 microcystin variants per strain. Thus, the results indicate that S1 and S5 had higher than mean values, and strains isolated during mid-summer (S2, S3 and S4) had lower than mean values. When the equality of the means was tested by a permutation test, the p-value was 0.006, i.e., the differences were clearly significant.

Among the 37 strains that were microcystin producers, between 1 and 12 microcystin variants were produced simultaneously (4.9 microcystins on average). Thirteen strains produced only one microcystin variant. Of these, 12 strains produced MC-RR and one strain produced MC-HilR. Microcystin profiles were strain-specific for the 24 strains that produced two or more microcystin variants, i.e., individual strains displayed unique microcystin chemotypes. Altogether, a total of 26 different *M. botrys* chemotypes were observed (Table 2). The variation includes which microcystin variants were produced, how many, in what combinations, as well as in what relative amounts (based on chromatogram peak area) they were produced (Appendix A).

A principal component analysis based on microcystin production indicated that three groups of strains clustered together, and that clustering was based on microcystin profiles, but also partly related to time of sampling (Figure 3). The first cluster (indicated in red) consisted of strains that produced the maximum number of microcystin variants observed in our data, all of which were sampled during sampling occasion S1. The second cluster (indicated in blue) consisted of “intermediate” strains, which produced 4–11 microcystin variants. All strains within this group were sampled on sampling occasion S4 and S5, except one strain (S1–79), which was sampled at S1. The third cluster (indicated in pink) consisted of strains that produced 1–2 microcystin variants, and included MC-producing strains sampled during all sampling occasions except S1. It is noteworthy that strains sampled within S2 and S3 were found in cluster 3. Overall, PC1 explained 51% of the variance, and PC2 explained 13%. The dendrogram of the Jaccard dissimilarity matrix of the microcystin variants present in the sampled strains supports the cluster structures indicated by the PCA (Figure 4).

Among the 18 microcystin variants in our dataset, MC-RR, [Dha^7^]MC-RR and MC-LR were the most common, produced by 92%, 57%, and 54% of strains respectively (Table 2, Appendix A). MC-RR was present in strains that were sampled during the whole period, whereas [Dha^7^]MC-RR and MC-LR were not present in strains sampled during the second and third sampling occasion. [Asp^3^]MC-RY, [Asp^3^]MC-RR and the unknown microcystin variants MC861 and MC509 were sampled only at S1 and S5, but [Asp^3^]MC-RR and MC509 were common and [Asp^3^]MC-RY and MC861 were rare within those sampling occasions. Some microcystins were rare in the whole data set. For example, MC-FR, [Ser^1^]MC-VR and the unknown variant MC502 occurred only during samplings S1, S4, and S5, respectively. Among the microcystins for which standards were available (MC-LR, MC-RR, [Asp^3^, Dhb^7^]MC-LR, [Dha^7^]MC-RR, and MC-YR), MC content was calculated to ng per mg dry weight (dw) of *M. botrys* cells (Appendix A). MC-LR had the highest mean content among these, and ranged between 402 and 473 ng mg^−1^ dw for strains sampled at the sampling points S1, S4, and S5. MC-RR, which was present in strains at all sampling points, had a mean content ranging between 38 and 117 ng mg^−1^ dw. For MC-YR, a high mean of 612 was obtained for S1 strains, while S3–S5 strain means ranged between 165 and 213 ng mg^−1^ dw.

Supporting background data on lake chemistry provided by the Vomb Water Works (Appendix A) showed that pH was stable (varying from 8.3 to 8.7) during the sampling period. Nitrate went from low to below detection by mid-July, while ammonium declined more slowly. Phosphate was lowest in early summer, and increased in late summer. The ratios of the inorganic soluble nutrients (N:P) indicated strong P-limitation in early summer, which switched to N-limitation in August and September, according to Redfield ratios.

## 3. Discussion

In this study we isolated an unprecedented high number (130) of *Microcystis botrys* strains throughout the duration of a bloom, and performed in-depth microcystin variant profiling of each isolate. We show that *M. botrys* strains sampled from this bloom displayed considerable variation in microcystin chemotypes, and that MC-producing *M. botrys* strains co-existed with non-MC-producing strains during the sampling season. Previous studies based on quantitative PCR targeted on single specific microcystin (*mcy*) genes, have shown that there is temporal and spatial variation in the distribution and dynamics of *mcy* genes. From this data it has been inferred that both MC-producing and non-MC-producing strains are found in several cyanobacterial genera, and that both types co-exist in natural populations (e.g., [29,30,38,44,45,46,47]). Indeed, by analyzing MC-production in the isolated strains, we verified that the *M. botrys* population in Lake Vomb also consisted of co-existing MC-producing and non-MC-producing strains. Moreover, we could detect a shift in the ratio of MC-producing and non-MC-producing strains over time, with MC-producing strains ranging from 8 % on July 14, to the maximum contribution of 52% on September 8. Although we isolated a large number of strains, we cannot entirely rule out that there was some isolation or culture bias. Observations from previous studies have shown that the prevalence of *mcy*-amplifying strains varies and that they can rapidly increase in occurrence. For example, the percentage of *mcy*-amplifying *Microcystis* sp. strains increased from 0.7% to 100% during a sampling in Lake Erie [29] and from 7.1% to 71% in Lake Wannsee (Germany) [48]. The results of the current study suggest that the increased ratio of MC-producing strains sampled at the end of the summer, may contribute to explaining the bulk microcystin peak concentrations observed repeatedly in several lakes in the region [49]. However, those peaks also coincided with high cell concentrations, and further studies are needed to establish whether the pattern from our study is repeated.

The sampled *M. botrys* strains displayed considerable variation in microcystin profiles throughout the sampling period. Among the 37 strains that produced microcystin, a total of 18 different microcystin variants were present in the population. Individual strains produced a maximum of 12 microcystins. This number must be placed in relation to the more than 240 microcystins that have been previously described in microcystin producing cyanobacteria [26]. For example, *Microcystis* strain CAWBG11 was found to produce 27 different microcystin variants [28], whereas *Microcystis* strains in other studies [50,51,52] produced less than five microcystins simultaneously. Additionally, a literature summary [28]) has reported a median number of 4–5 microcystins produced simultaneously by cyanobacterial strains. Moreover, cyanobacteria are able to produce several oligopeptides with structural variance, such as cyanopeptolines, anabaenopeptins, nodularins, and aeruginosins simultaneously [53,54,55,56,57,58]. In the current study, strains from the sampled *M. botrys* population separated into a total of 26 chemotypes based on microcystin variants (Table 2 and Figure 4). The majority of MC-producing strains thus displayed unique profiles, and these individual chemotypes were sampled only once during the bloom. Interestingly, the 12 strains that produced only the microcystin MC-RR re-occurred throughout the whole season. Although these 12 strains represent the same microcystin chemotype, based on the available data we cannot determine whether the strains belonged to a single or multiple genotypes. However, MC-RR production varied among the strains, ranging from 5 ng mg^−1^ dw in an S3 strain to 122 ng mg^−1^ dw in a S1 strain when grown in the same laboratory conditions, suggesting that there are strain differences. While some microcystin variants were common in strains throughout the whole bloom, others were either rare overall (e.g., [Ser^1^]MC-VR) or specific to strains sampled at a certain time period (e.g., [Asp^3^]MC-RR). This indicates that it is both the rare microcystins and the different combinations of microcystin variants that contribute to the observed variation in microcystin profiles.

Not only the composition, but also the content of the MC variants varied among strains. While MC-RR was the most common variant, MC-LR content was higher, when present, with a mean content of 426 ng mg^−1^ dw compared to 73 ng mg^−1^ dw for MC-RR. From a total microcystin perspective, this indicates that MC-LR is perhaps more important than MC-RR in this population of *M. botrys*. On the other hand, MC-YR was also very high in S1 strains (mean 612 ng mg^−1^ dw), and for most other microcystin variants no standards were available. However, here we cannot determine contributions of each toxin in the field, nor the toxicity of each variant to other organisms.

The high diversity in chemotypes observed in this study, with up to 12 microcystin variants, and 26 different combinations among MC-producing strains, raises the question why there is such a high diversity. A plausible scenario is that microcystin chemotype composition could be subjected to selective forces. It is possible that *M. botrys* with particular chemotypes, or specific microcystin variants, could be favored during different environmental conditions. In that case, we would expect that different chemotypes would be more prevalent during certain times of the year, or in lakes where these environmental criteria are optimal. Although the majority of individual chemotypes occurred only once in our dataset, we did observe that strains producing several microcystin variants occurred in early and late summer (S1 and S5), whereas strains that produced few microcystins occurred in the middle of the summer (S2, S3 and S4). Irrespective of the potential ecological functions of microcystin, the question of why there are so many structural variants remains. If microcystins do have a protective function against grazers and/or pathogens [59,60], one explanation could be that cyanobacteria are exposed to multiple enemies, which could potentially select for multiple peptide variants. However, the evolution of microcystin synthetase genes pre-dates the occurrence of metazoans [32]. In fact, a transcriptome study showed that genes coding for putative grazer deterring compounds (such as microcystin), were largely unaffected when *Microcystis* was exposed to predators [61]. On the other hand, field studies have shown that microcystin concentrations are correlated to zooplankton composition, where high concentrations appear detrimental to potential grazers [49,62]. An alternative explanation to the existence of multiple structural variants is that large variation allows for rapid adaptation to sudden environmental changes, such as nutrient pulses. Many studies have shown that microcystins are involved in the cell metabolism (e.g., [35,63,64]), which supports this explanation. However, to date there is no information regarding the role of different MC variants. While there are several plausible explanations to why cyanobacterial compounds are produced, there is no consensus. Experimental results are often very strain specific, and in most monoculture and co-culture experiments, only one MC-producing and one non-MC-producing strain, are used. Hence, it is difficult to extrapolate the results of laboratory studies to natural populations.

The co-existence of both MC-producing and non-MC-producing strains within *Microcystis* blooms suggests that microcystin production has a cost, as shown in several studies [44,65,66], and that certain environmental conditions favor one, but not the other. In fact, the majority of strains isolated in this study were not MC-producers, indicating that microcystin-production could be beneficial during certain conditions, but not always. Several abiotic (e.g., light, nutrient availability, temperature) and biotic factors (e.g., intra- and interspecific competition, predation) have been suggested to influence shifts in abundance of MC-producing and non-MC-producing strains [29,30,47,67,68,69], but the results are somewhat conflicting. In part this may be due to the fact that several of these studies were based on quantitative PCR targeted at part of the *mcy* gene complex, rather than the observation of actual toxin producing strains (but see [51]). In controlled laboratory experiments, Kardinaal et al. [67] showed that non-MC-producing *Microcystis* strains performed better than MC-producing strains under light limitation, whereas Leblanc Renaud et al. [70] showed that MC-producing strains were dominant under both low and high light intensities. Furthermore, it has been suggested that production of microcystin seems to be beneficial under suboptimal growth conditions, but costly (in terms of strain/population growth) when conditions are favorable for cyanobacterial growth (see e.g., [44]). In the current study, we observed that the increase in MC-producing phenotypes coincided with a shift in the dissolved inorganic nutrient quota, from phosphorus-limited to nitrogen-limited between the end of August to September (Appendix A), suggesting that MC-producing strains are favored over non-MC-producing strains under nitrogen-limiting conditions. This pattern is in line with the experimental results by Suominen et al. [71], in which a MC-producing strain performed better than the non-MC-producing mutant under nitrogen limitation. Field studies based on qPCR also indicate that MC-producing strains perform better under low N:P ratios [29,69,72]. In addition, recent studies [72,73,74] suggest that the nitrogen species, such as NO_3_:NH_3_ ratios [72], might play a key role both as drivers of cyanobacterial blooms, and in the production of secondary metabolites such as microcystin.

To conclude, here we show that through a large scale effort in isolating *M. botrys* strains from a bloom, we could reveal that microcystin chemotype diversity within a population is very high. We could also demonstrate a variation not only in the proportion of MC-producing and non-MC-producing strains, but also a variation of microcystin variants produced, throughout the bloom duration. For future studies it would be useful to perform high resolution sampling of environmental data, as well as of proportions of MC-producing and non-MC-producing strains, together with genotyping of strains.

## 4. Methods

### 4.1. Sampling and Isolation of Microcystis Colonies

The cyanobacterial bloom in Lake Vomb, southern Sweden (55°41′44.0″N 13°35′41.0″E), was sampled on five occasions from June to September 2014. Sampling of the cyanobacterial community was done by concentrating surface water (250 mL) with a plankton net from the shore (mesh size 20 µm, max sample depth 1 m). Surface water was also collected in 1 L plastic bottles. Samples were stored in cooling bags with ice packs while transported to lab. In the lab, the lake water was immediately filtered through Whatman GF/C filters (GE Healthcare Life Sciences, Chicago, IL, USA), and then re-filtered through cellulose acetate sterile filters (VWR, Radnor, PA, USA), to use during isolation and for preparation of growth medium. *M. botrys* colonies were isolated from the cyanobacteria community samples on the same day as, or occasionally up to three days after, sampling. Biomass from the plankton net samples were screened in the microscope at 20–40 magnification (Eclipse TS100, Nikon, Tokyo, Japan) and individual *Microcystis botrys* colonies were picked out by micropipetting using disposable glass capillaries (Hirschmann Laborgeräte, Eberstadt, Baden-Württemberg, Germany). During isolation, colonies were stepwise washed three times in sterile-filtered lake water and visually inspected using microscope in order to eliminate presence of non-target organisms. Colonies were transferred to 150 µL growth medium in sterile cell culture 96-well plates (VWR, Radnor, PA, USA). At least 140 individual colonies were isolated at each sampling occasion. Initially, six different combinations of growth medium were tested: 100% modified MWC ([75], modified by adding 0.002 g L^−1^ Na_2_SeO_3_·5H_2_O, hereafter MWC+Se); 100% Z8 [76]; 50% MWC+Se + 50% sterile-filtered lake water; 50% Z8 + 50% sterile-filtered lake water; 25% MWC+Se + 75% sterile-filtered lake water; 25% Z8 + 75% sterile-filtered lake water. However, colonies growing in MWC+Se had greater survival (data not shown) and further use of Z8 was discontinued. Isolates were maintained in climate rooms under controlled conditions (20 °C temperature; initially at 15 µmol photons m^−2^s^−1^, after established growth in 20 µmol photons m^−2^s^−1^; 14:10 light:dark photoperiod). Colonies were regularly monitored for growth, performance, and survival, to check for contaminants and to re-isolate when needed. Colonies that divided and multiplied were transferred to 25 mL plastic culture flasks (Thermo Fisher Scientific, Waltham, MA, USA) for further cultivation under the same conditions as the cell culture plates (20 °C temperature; 20 µmol photons m^−2^s^−1^; 14:10 light:dark photoperiod). Established cultures were subsequently screened again microscopically to ascertain species identity and lack of contamination.

### 4.2. Harvest of Microcystis botrys Strains

*M. botrys* strains were harvested for microcystin analysis approximately one year after initial isolation and successful establishment. A total of 130 *M. botrys* strains, made up of at least 20 strains from each sampling occasion (S1–S5), were cultivated in 125 mL MWC+Se in climate rooms under growth conditions described above. When cultures had reached late exponential phase, approximately four weeks after inoculation to fresh growth medium (MWC+Se), they were harvested. Cultures were transferred to two 50 mL centrifuge tubes (VWR, Radnor, PA, United States) and centrifuged (Allegra X-30R Centrifuge, Beckman Coulter, Brea, CA, USA) for 10 min at 3000 g and 4 °C. The supernatant was discarded and pellets were washed and re-dissolved with 4 °C MWC+Se, thereafter centrifuged again. This was repeated at least twice. The resultant pellets were again washed and re-dissolved in 1 mL 4 °C MWC+Se, transferred to 1.5 mL Eppendorf-tubes (VWR) and centrifuged (Microfuge 22R Centrifuge, Beckman Coulter, Brea, CA, USA) for 10 min at 3000 g in room temperature. Supernatant was removed and the final pellets were stored in −80 °C.

### 4.3. Extraction and LC-MS/MS Analysis

Lyophilized cyanobacterial biomass (5 mg) was extracted in 75% methanol in water (1 mL, LC-MS grade) by 5 min vortexing followed by 1 min sonication in ultrasonic bath. After centrifugation, the supernatant was transferred to chromatographic vials and analysed by LC-MS/MS. The chromatograph (1200 SL series, Agilent Technology, Santa Clara, CA, United States) was interfaced with a mass spectrometer (QTrap 5500, AB Sciex, Concord, ON, Canada) via a turbo ion source (550 C; 5.5 kV). Chromatographic separation of 5-µL samples was performed on a Zorbax Eclipse XDB-C18 column (4.6 mm, 150 mm, 5 µm) (Agilent Technology, Santa Clara, CA, USA) maintained at 40°C. Compounds were eluted at a flow rate of 0.6 mL min^−1^ with a mobile phase composed of 5% acetonitrile in Milli-Q water (A) and acetonitrile (B), both containing 0.1% (*v/v*) formic acid. The content of mobile phase B changed from the initial 15% to 99% in 8 min and was then held at 99% for 10 min. The tandem mass spectrometer was operated in positive ionization mode, with curtain gas (N_2_) pressure of 20 PSI and declustering potential of 60 V. The cyanopeptide profile was determined in information-dependent acquisition mode (IDA). For ions with *m/z* in a range 500–1100 and signal above the threshold of 500,000 cps, fragmentation spectra were collected at a collision energy of 60 V. Structure elucidation was performed based on a fragmentation spectrum of pseudomolecular ions. The unknown compounds with *m/z* at 1035, 1031, 861, 528, 509, and 502 were classified as MCs based on the presence of several diagnostic ions in their spectra. For MCs whose standards were available (Alexis Biochemicals, San Diego, CA, USA: MC-LR, MC-RR, [Asp^3^, Dhb^7^]MC-LR, [Dha^7^]MC-RR, and MC-YR), the level of detection was between 0.1 and 0.5 ng mL^−1^. The concentrations (MC ng mg^−1^ dw) of cyanopeptides in extracts were calculated for MCs where standards were available. For MCs where no standards were available, the relative amounts of the cyanopeptides in extracts were estimated based on the peak area in the extracted ion chromatogram in relation to biomass.

### 4.4. Statistical Analysis

To test for differences in temporal distribution between MC-producing and non-MC-producing *M. botrys* strains, a chi-square goodness-of-fit test was performed in SPSS [77]. A principal component analysis (PCA) was performed to search for patterns in the MC profiles among MC-producing *M. botrys* strains. All detected MC variants (*n* = 18) and the relative amounts (based on chromatogram peak area) produced by each strain were included in a factor analysis, which was performed in SPSS using covariance matrix. The factor scores were saved as variables using a regression method and plotted with R [78]. MC profiles of toxigenic strains were visualized by hierarchical clustering using a Jaccard dissimilarity matrix (presence/absence, see Table 2) [79]. A permutation test was performed to test for equality of means of the number of MC variants produced among the sampling occasions. In each simulation cycle the individual strains were randomly assigned without replacement to the sampling occasions, keeping the number of strains fixed, i.e., the same numbers of MC-producing strains that were actually observed in the study. For each cycle the variance among the sampling occasions were calculated and compared to the observed variance. Similarly, a permutation test was performed for the proportion of strains that produce MCs. In each cycle, the 37 MC-producing strains were randomly assigned among the five sampling occasions, taking the total number of investigated strains into account. In each run we calculated two statistics from the simulated data. The first was the variance among proportions which was then compared to the observed variance. Secondly, in order to test for linear trend, we calculated the covariance between the sampling day and the proportion and then compared that value to the observed covariance. For MCs where standards were available (MC-LR, MC-RR, [Asp^3^, Dhb^7^]MC-LR, [Dha^7^]MC-RR, and MC-YR) the mean values and standard deviations were calculated for each sampling occasion (S1–S5) and plotted in KaleidaGraph (ver. 4.5) [80].

### 4.5. Water Chemistry

Background data for water chemistry parameters was provided by Vomb Water Works. Water samples were taken on a monthly basis at the water inlet pump station of the Vomb Water Works. The parameters included here were turbidity, pH, conductivity, ammonium, nitrite, nitrate, ortho phosphate, and iron. Iron was measured by inductively coupled plasma mass spectrometry (ICP-MS) (ISO 17294-2:2016), phosphate by the ammonium molybdate spectrometric method (ISO 6878:2004), ammonium by flow analysis and spectrometric detection (ISO 11732:2005), and nitrite by cadmium reduction. N:P-ratios were calculated in this study based on the dissolved organic nutrients. Water chemistry data for samples taken in May through September 2014 and are included as background data (Appendix A), but not used in the statistical analysis.

## Figures and Tables

**Figure 1 toxins-11-00698-f001:**
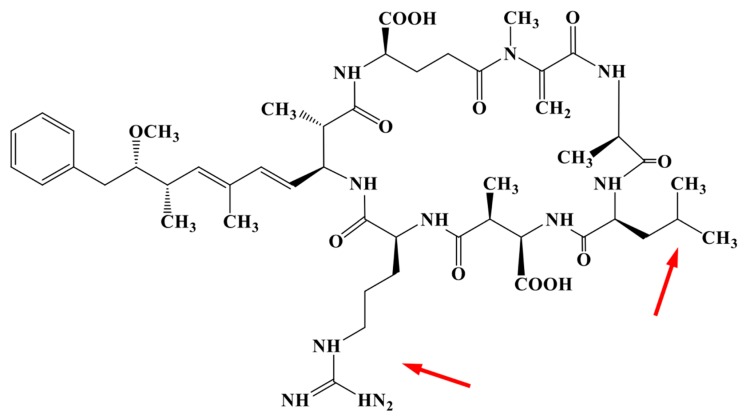
General structure of microcystin, illustrating the common variant MC-LR. Arrows indicate the variable amino acids in positions 2 and 4 in the cyclic structure [14,21,22].

**Figure 2 toxins-11-00698-f002:**
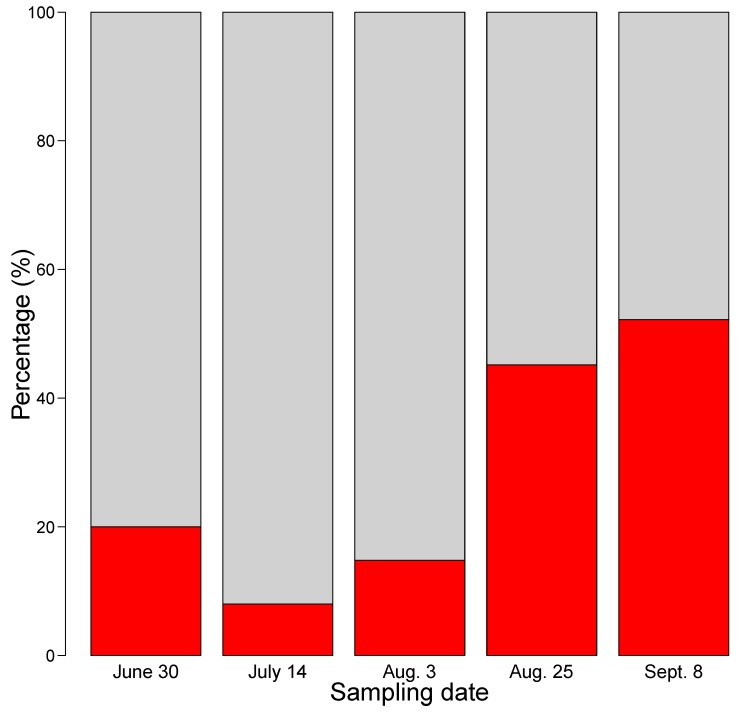
Barplot showing the percentage (%) of strains isolated from Lake Vomb, that were MC-producers (red) and non-MC-producers (grey).

**Figure 3 toxins-11-00698-f003:**
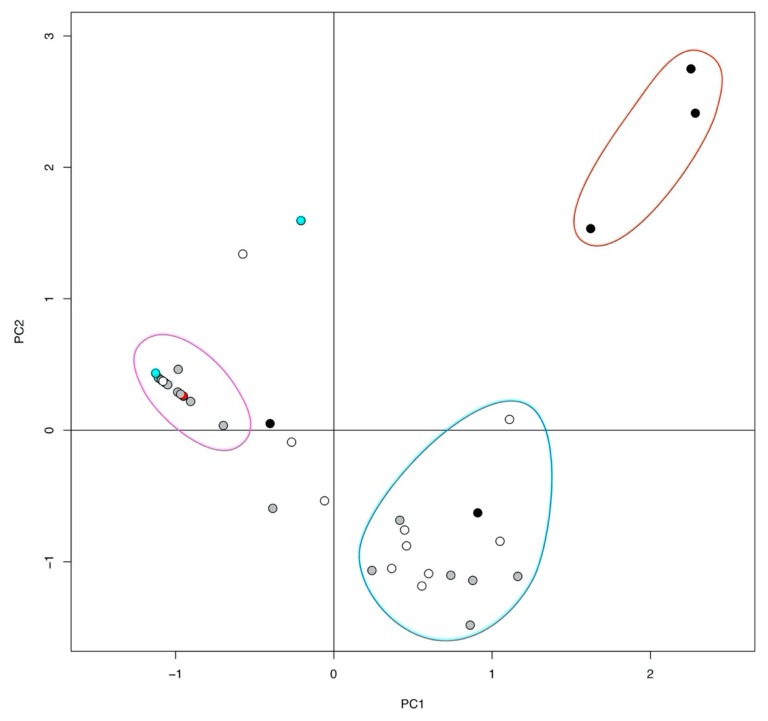
Clustering based on a PCA of the 37 microcystin (MC) producing *Microcystis botrys* strains (filled circles) sampled during the summer bloom. The data analyzed includes microcystin variants and relative amounts produced. Colors denote sampling date (prefix S1–S5): black = S1, red = S2, blue = S3, grey = S4, white = S5. Circled clusters refer to main clusters observed, which relate to number of MC variants, red = cluster 1 (≥12 MC variants), blue = cluster 2 (4–11 MC variants), pink = cluster 3 (1–2 MC variants).

**Figure 4 toxins-11-00698-f004:**
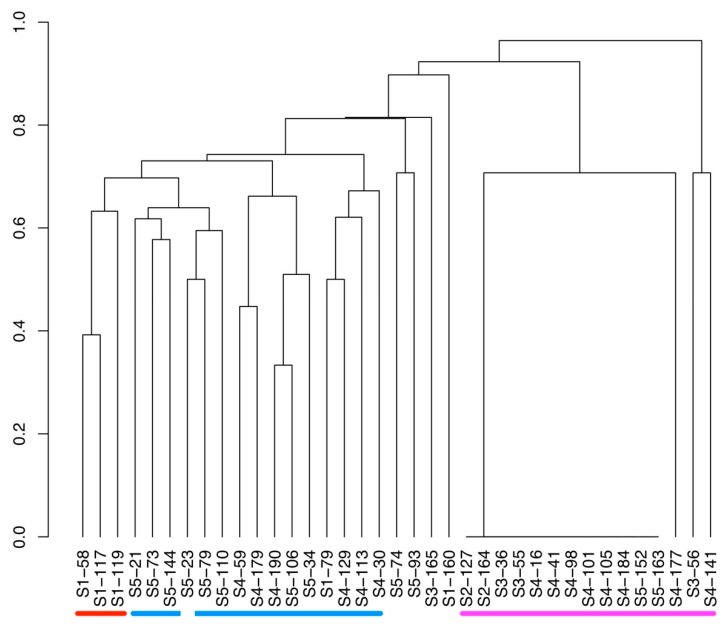
Dendrogram of the Jaccard’s dissimilarity matrix based on presence/absence of microcystin variants in the analyzed *Microcystis botrys* strains. Colors refer to clusters in Figure 2, red = cluster 1 (≤12 MC variants), blue = cluster 2 (4-11 MC variants), pink = cluster 3 (1-2 MC variants).

**Table 1 toxins-11-00698-t001:** *Microcystis botrys* sampling, isolation, and survival. Strain prefix is based on sampling occasion.

Strain Prefix	Sampling Date	Water Temp at 0.5 m Depth (°C)	Colonies Isolated	Cultures Established	Cultures Analyzed	Number of MC Producing Strains
*S1*	2014-06-30	19	143	35	25	5
*S2*	2014-07-14	20	192	33	25	2
*S3*	2014-08-03	26	192	35	27	4
*S4*	2014-08-25	18	148	35	30	14
*S5*	2014-09-08	19	168	30	23	12

**Table 2 toxins-11-00698-t002:** Microcystin profiles for MC-producing strains sampled in Lake Vomb during summer 2014 based on presence/absence of microcystin variants in all strains. Dot denotes presence. Unknown microcystin variants are denoted MC? and the *m/z* values of their pseudomolecular ions are given.

Sampling Date (2014)	Toxigenic Strain ID	Microcystin Variants
MC-WR	[Asp^3^]MC-ThTyrR	MC-RY	MC-RR	MC?	[Asp^3^] MC-RY	MC-FR	[Asp^3^] MC-RR	[Dha^7^] MC-RR	MC-HilR	[Ser^1^] MC-VR	MC-LR	[Asp^3^Dhb^7^] MC-LR	MC?	MC?	MC?	MC?	MC?
*m/z* [M+H]^+^						1035									861	528	1031	509	502
June 30	S1-58	●	●	●	●	●		●	●	●			●	●	●			●	
S1-79	●		●	●					●	●		●			●			
S1-117	●	●	●	●		●	●	●	●			●	●	●			●	
S1-119	●	●	●	●			●		●	●		●		●	●	●	●	
S1-160	●							●	●									
July 14	S2-127				●														
S2-164				●														
August 3	S3-36				●														
S3-55				●														
S3-56										●								
S3-165		●	●	●												●		
August 25	S4-16				●														
S4-30			●	●					●			●						
S4-41				●														
S4-59			●	●	●				●		●	●	●		●	●		
S4-98				●														
S4-101				●														
S4-105				●														
S4-113	●			●					●			●				●		
S4-129	●		●	●					●			●			●	●		
S4-141										●		●						
S4-177				●								●						
S4-179		●	●	●	●				●			●	●		●	●		
S4-184				●														
S4-190		●	●	●					●	●		●	●			●		
September 8	S5-21			●	●				●	●			●			●		●	
S5-23		●	●	●				●	●			●				●	●	
S5-34		●	●	●				●	●	●		●	●					
S5-73	●	●	●	●				●	●			●			●	●	●	●
S5-74				●					●	●			●			●		
S5-79		●		●				●	●			●					●	
S5-93		●		●					●	●								
S5-106		●	●	●					●	●		●	●	●		●		
S5-110		●	●	●					●	●		●					●	
S5-144	●		●	●		●		●	●			●				●	●	
S5-152				●														
S5-163				●

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
