# Peer review of "High Diversity of Microcystin Chemotypes within a Summer Bloom of the Cyanobacterium Microcystis botrys"

_toxins, 2019, doi:10.3390/toxins11120698_

Round 1
Reviewer 1 Report
The paper presents some interesting data on the variability of Microcystis botrys in relation to the production of microcystins variants.
This species is quite diffuse in freshwater basins, especially in freshwater systems in Northern Europe, but there is not much information on its ecological requirements and/or its toxicity, an important aspect for water quality and its safety. Most of the information is indeed about M. aeruginosa, or Microcystis sp. Therefore new data are needed.
The Authors made a lot of work in isolating strains during a few months bloom, and eventually managed to culture 168 strains from 5 data sampling, out of > 800 colonies isolated. Of these cultured strains, they examined the production of microcystins in 130 colonies (why not all of them?) and found that most of the colonies do not produce microcystins at all, and a large variability in the number of variants produced and the combinations of the variants in each colony in the toxic colonies. It’s not a new result, but it’s the first time to my knowledge that it is done on M. botrys. And the paper is well written.
However, these data are only qualitative, are not related to any specific variable in the field and they alone cannot give any insight “in the variation of microcystin chemotypes during a bloom” (Introduction row 78).
Furthermore, are based on a small percentage of isolated colonies. The Authors aknowledged that these are results from lab cultures, but are confident that the number of isolates is high enough to get reliable data. To me these data are enough to show the “potential” variability of M. botrys colonies in producing microcystins, but are not enough to support any other speculation on the reasons of this variability. It is also a pity that the Authors did not give any quantitative data on cell quota of microcystins, that can be very useful in characterization of the species and on the assessment of the risk in case of a bloom. More data would be needed to discuss the results presented, that can be genetic (it is possible that all strains have the mcy gene, but for some other reason are not producing microcystins), or related to what was going on during the bloom in the field: the profile of microcystins found during the bloom, the quantity of microcystins found and possible relation with the number of toxic genotypes present in the field, the size of the colonies (number of cells), and so on.
The discussion being based on these few data, remains too general and does not address any specific problem. Correlating the change in number of toxic strains to “the end of summer” is too general and does not give new information (rows 170-). Also the discussion on the reasons why there are so many variants of microcystins is too general and it is not based on your data.
I would suggest to present the data as a short communication, not to waste the important lab work and the data, that as I said are missing for M. botrys.
I don’t have specific comments.
Reviewer 2 Report
The manuscript is well written with clear objectives and supporting results. The quality of work warrants its acceptance but with some minor modifications:
Since the whole manuscript is about microcystins. A figure with a chemical structure would be of great help to readers and the brief importance of studying MCs. No mention about how the purity of cultures was checked after culturing. Was each sample analyzed in replicates? Why do authors need pure cultures for this study as in nature blue-green are seldom found without associated microbes and MS these days are sensitive enough to pick up MCs out of complex mixtures?Author Response
Please see the attachment.

Round 2
Reviewer 1 Report
As I said, I think the data set provided is very valuable, it is the result of a very important work and add information on Microcystis botrys, for which there are not many data. The few adjustments on the discussion make it better and acceptable for publication, even if I have some comments.
It is unclear to us which data the Reviewer considers qualitative. Although the results are not expressed in mg MCs/g dw cyanobacteria, all data presented in the study are quantitative (peak height area for microcystins, allowing comparison between strains), and so are the environmental data (Supplementary Table S1).
I still think that data on MC are qualitative. I’m not saying they are not extensive, but the chromatograms area, as the Author says, only allows comparison between those cultured strains, for this specific research. Since a lot of work has been done, the expression of MC data as ug/L or ug/biomass or biovolume for the known MC variants would allow comparisons with other studies.
We do agree thought that we do not have enough data to explain the reason for the observed variability in MC/non-MC-producing strains. Our intention was to point out in the manuscript that environmental factors possibly could affect this variability, but that this is only a hypothesis that we raise in the discussion, and which warrants further study. We have attempted to clarify this point in the manuscript, see page 9, lines 536-538.
Unfortunately there must be a mismatch in the number of lines listed in the cover letter with my pdf version, and I don’t know at what part the Author refers. I agree with the modification at page 9, lines 188-190. I also agree with the lines 245-247 at page 10, but I don’t understand the following phrase: While convenient in many ways, the qPCR methodology suffers from issues with false positives and negatives, and in fact measures the presence of the genes, but not whether microcystins are produced.
Maybe “in fact” can be “futhermore”? Or with “false positives and negatives” the Author refers to the production of MC? It is not clear, and I would simply cancel the phrase.
Last page 10 rows 260, the phrase “Thus, we propose the hypothesis that late summer conditions likely select for the MC-producing phenotypes with the most-adapted metabolite profile/chemotype in this ecosystem” does not add anything to the discussion. It is much better the Author refers to N:P ratio and nutrient limiting conditions (environmental factors), as he/she does in the rest of the discussion, than to “summer conditions” (that cannot be considered an environmental factor). I would cancel the phrase.
